# Regulation of Glutathione *S*-Transferase Omega 1 Mediated by Cysteine Residues Sensing the Redox Environment

**DOI:** 10.3390/ijms25105279

**Published:** 2024-05-12

**Authors:** Kwonyoung Kim, Jeongin Choi, Sana Iram, Jihoe Kim

**Affiliations:** Department of Medical Biotechnology, Yeungnam University, Gyeongsan 38541, Republic of Korea; cathykimva98@ynu.ac.kr (K.K.); jeongin@yu.ac.kr (J.C.)

**Keywords:** glutathione *S*-transferase omega 1, protein deglutathionylation, redox regulation, cysteine, pathological mutation

## Abstract

Glutathione *S*-transferase omega 1 (GstO1) catalyzes deglutathionylation and plays an important role in the protein glutathionylation cycle in cells. GstO1 contains four conserved cysteine residues (C32, C90, C191, C236) found to be mutated in patients with associated diseases. In this study, we investigated the effects of cysteine mutations on the structure and function of GstO1 under different redox conditions. Wild-type GstO1 (WT) was highly sensitive to hydrogen peroxide (H_2_O_2_), which caused precipitation and denaturation at a physiological temperature. However, glutathione efficiently inhibited the H_2_O_2_-induced denaturation of GstO1. Cysteine mutants C32A and C236A exhibited redox-dependent stabilities and enzyme activities significantly different from those of WT. These results indicate that C32 and C236 play critical roles in GstO1 regulation by sensing redox environments and explain the pathological effect of cysteine mutations found in patients with associated diseases.

## 1. Introduction

Glutathione *S*-transferase omega 1 (GstO1), an omega-class glutathione transferase, is structurally and functionally distinct from glutathione transferases of other classes [1]. Most glutathione transferases are involved in phase II detoxification via glutathione conjugation, but the physiological function of GstO1 is not completely understood. GstO1 accepts various substrates including dehydroascorbate, monomethylarsonate (V), and phenylacetyl glutathione, and catalyzes their reductions [2,3,4]. In addition, GstO1 catalyzes protein deglutathionylation, which removes glutathione (GSH) from glutathionylated proteins with cysteine residues conjugated to glutathione [5]. This deglutathionylation activity suggests that GstO1 participates in the protein glutathionylation cycle, regulating protein functions and structures in different redox environments [6].

GstO1 is closely associated with human diseases such as cancers, neurological diseases, inflammation, and obesity [7]. In particular, GstO1 is highly expressed in cancer cells and is involved in regulating signaling pathways for cancer development [8,9,10]. Therefore, GstO1 has been an attractive target for cancer treatment. In a previous study, a GstO1 inhibitor was developed for colorectal cancer treatment [8]. The C1-27 inhibitor was highly specific for GstO1 and inhibited the enzyme activity by covalent modification of the catalytic cysteine residue (C32). Furthermore, C1-27 was highly cytotoxic to different types of cancer cells and effectively suppressed the proliferation of colon cancer cells in vitro and in vivo. Other studies have reported that GstO1 plays a pro-inflammatory role and modulates the metabolism of activated macrophages through the Lipopolysaccharide (LPS)- Toll-like receptor 4 (TLR-4) pathway [11,12]. The inhibition of GstO1 suppressed LPS-induced inflammation, and the high-fat-diet-induced obesity was significantly reduced in the GstO1 knock-out mice [12]. Recently, we also reported that GstO1 inhibition suppresses adipocyte differentiation by decreasing protein deglutathionylation, which indicates that GstO-catalyzed deglutathionylation is important in adipocyte differentiation [13].

GstO1 contains a catalytic cysteine residue (C32) that is highly conserved in mammals including humans [8]. In addition to the catalytic cysteine, GstO1 contains three additional conserved cysteine residues (C90, C191, and C236) (Figure 1A). Single nucleotide polymorphism (SNP) analyses of patients with different types of diseases found cysteine mutations, in particular, of C32 and C236, which could be pathological under disease conditions [14,15,16,17]. In this study, we investigated the effects of mutations of conserved cysteines on the enzyme activity and the stability of GstO1. Wild-type GstO1 (WT) was found to be highly sensitive to redox changes. Hydrogen peroxide (H_2_O_2_) caused the rapid precipitation and denaturation of GstO1 at a physiological temperature, and these were efficiently inhibited by glutathione. Cysteine mutants C32A and C236A exhibited significantly different redox-dependent stabilities and enzyme activities. Our results indicate that C32 and C236 play a critical role in the regulation of GstO1 by sensing the redox environment and explain pathological cysteine mutations.

## 2. Results

### 2.1. Preparation of Cysteine Mutant GstO1 and Characterization

GstO1 contains four highly conserved cysteine residues including the catalytic cysteine (Figure 1A). The cysteine residue of mouse GstO1 was mutated to alanine by site-directed mutagenesis. Wild-type and cysteine mutant GstO1 (WT, C32A, C90A, C191A, and C236A) were homogenously prepared, as described in the materials and methods (Figure 1B). SDS-PAGE showed molecular weights of ~30 kDa for mutant GstO1 and WT. However, non-reduced SDS-PAGE showed separation patterns different between WT and mutant GstO1 (C90A, C191A, and C236A), except for C32A, suggesting that the conformations of the mutant GstO1 could be different from the WT conformation (Figure 1C).

Enzyme activities were determined for WT and mutant GstO1 (Table 1). WT had a thiol transferase activity of 116 ± 11 mU/mg, a deglutathionylation activity of 3.0 ± 0.1 mU/mg, a DHA reductase activity of 132 ± 31 mU/mg, and a 4-NPG reductase activity of 9.1 ± 0.2 U/mg. The catalytic cysteine mutant C32A did not exhibit these enzyme activities, but showed a low glutathione *S*-transferase activity of 40 ± 4 mU/mg, consistent with a previous observation [8]. The mutation of other conserved cysteine residues also affected and altered enzyme activities to varying degrees. The most significant change was observed for C236A, which showed increases in the thiol transferase activity (150 ± 22 mU/mg), deglutathionylation activity (3.5 ± 0.1 mU/mg), and the DHA reductase activity (166 ± 19 mU/mg). In addition, C236A exhibited a 4-NPG reductase activity of 6.1 ± 0.6 U/mg, which was significantly lower than that of WT.

**Figure 1 ijms-25-05279-f001:**
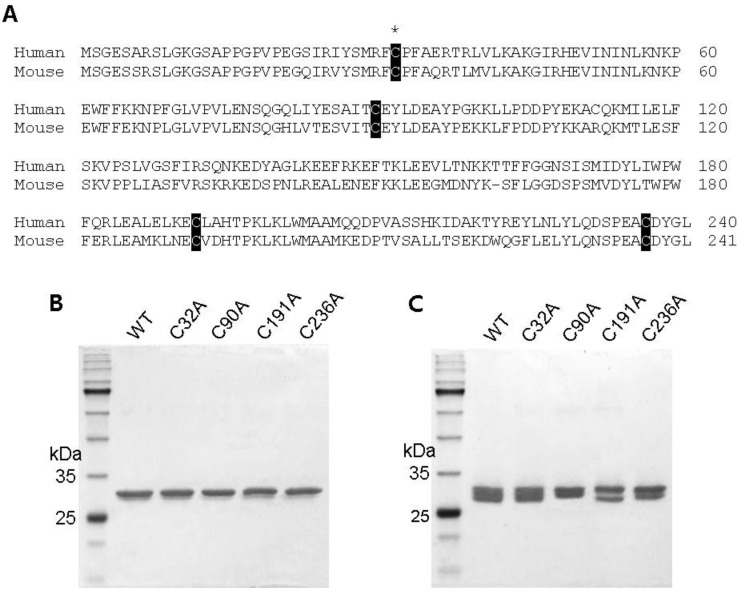
Conserved cysteine residues in GstO1. (**A**) Amino acid sequence alignment for human and mouse GstO1. Conserved cysteine residues are indicated by white letters on a black background, and the catalytic cysteine (C32) is indicated by an asterisk. (**B**,**C**) SDS-PAGE analysis for purified wild-type (WT) and cysteine mutant GstO1 under reducing (**B**) and non-reducing conditions (**C**).

### 2.2. Changes in GstO1 Stability Dependent on the Redox Environment

WT GstO1 was stable at room temperature and showed no significant change in its solubility or enzyme activities. However, hydrogen peroxide (H_2_O_2_) destabilized WT GstO1 and showed rapid precipitation in the presence of H_2_O_2_ (Figure 2A). The stability of WT GstO1 was examined at different temperatures by differential scanning fluorimetry (DSF), and *T*_m_ = 60.0 ± 0.3 °C was determined (Figure 2B and Figure 3A). In the presence of H_2_O_2_, the initial fluorescence at 20 °C was significantly high and fluctuated by increasing temperature. This denaturation curve indicated that the *T*_m_ of WT GstO1 was dramatically decreased to <20 °C, which also indicated H_2_O_2_-induced destabilization (Figure 2B and Figure 3B). Glutathione (GSH) alone insignificantly changed the *T*_m_ of WT GstO1, and a *T*_m_ = 59.0 ± 0.2 was determined in the presence of GSH (Figure 3C). But, in the presence of H_2_O_2_ and GSH, *T*_m_ = 61.0 ± 0.8 °C was determined, indicating that GSH inhibits the H_2_O_2_-induced destabilization of WT GstO1 (Figure 3D). The stability of WT GstO1 was also measured at the constant temperature of 37 °C in the presence of different H_2_O_2_ concentrations by isothermal denaturation (ITD) (Figure 2C). Denaturation rates increased by increasing H_2_O_2_ concentrations and resulted in EC_50_ = 0.20 ± 0.04 mM (Figure 2D). The denaturation rate was significantly decreased in the presence of additional GSH with 1.0 mM H_2_O_2_ (Figure 4D).

### 2.3. Cysteine Residues Critical for the Redox-Dependent Stability of GstO1

The thermal stabilities of cysteine mutants were examined by DSF (Figure 3), and estimated *T*_m_ values are summarized in Table 2. The *T*_m_ values for C32A, C90A, and C191A were estimated to be 45.0 ± 0.3 °C, 57.0 ± 1.5 °C, and 56.0 ± 0.7 °C, respectively, and significantly lower than that of WT GstO1 (Figure 3A), while C236A showed an insignificant change in *T*_m_ (61.0 ± 0.2 °C) and was similar to that of WT GstO1. In the presence of H_2_O_2_, C90A and C191A were highly unstable like WT and showed *T*_m_ decreasing to <20 °C (Figure 3B). C32A was also destabilized by H_2_O_2_ and showed a small decrease in *T*_m_ (41.0 ± 0.8 °C), but C236A was not significantly affected. GSH alone caused relatively small changes in *T*_m_ for WT and most cysteine mutants, while C32A showed a significant *T*_m_ increase of ~7 °C in the presence of GSH (Figure 3C). However, in the presence of H_2_O_2_, GSH significantly stabilized C32A, C90A, and C191A and increased their *T*_m_ values (Figure 3D). On the other hand, GSH did not change the *T*_m_ of C236A in the presence of H_2_O_2_.

The stabilities of mutant GstO1 were also examined by ITD (Figure 4). Incubation at 37 °C did not cause a significant denaturation of WT, C90A, C191A, or C236A, but C32A was rapidly denatured with a half denaturation time of ~20 min (Figure 4A). The denaturation of C32A was faster in the presence of H_2_O_2_ (a half denaturation time of ~15 min) (Figure 4B). Other cysteine mutants also exhibited rapid denaturation in the presence of H_2_O_2_, whereas C236A remained stable without significant denaturation. GSH stabilized C32A and decreased the denaturation rates in the absence and presence of H_2_O_2_ (Figure 4C,D). Other cysteine mutants C90A and C191A were also stabilized by GSH in the presence of H_2_O_2_, showing decreases in the denaturation rates (Figure 4D).

### 2.4. Redox-Dependent Changes in the Conformation of GstO1

After incubation at 37 °C, WT and cysteine mutants were analyzed by non-reduced SDS-PAGE (Figure 5). WT exhibited different separation patterns in the presence of H_2_O_2_ or GSH, indicating the redox-dependent conformational changes (Figure 5A). Cysteine mutants also showed different separation patterns in the presence of H_2_O_2_ or GSH (Figure 5A). In the presence of H_2_O_2_, oligomeric forms of WT and the cysteine mutants C32A, C90A, and C191A were detected at high molecular weights (>60 kDa) that increased with incubation time (Figure 5A,B). These oligomer generations were inhibited by GSH, indicating that they are produced by disulfide bond formation during denaturation. However, the C236A oligomer was not detected in all conditions (Figure 5A), which consistently indicated that C236A was resistant to H_2_O_2_-induced denaturation.

### 2.5. Cysteine Residues Critical for the Deglutathionylation Activity of GstO1

In a previous study, we found that GstO1 expression was elevated during the early phase of adipocyte differentiation and accompanied by the deglutathionylation of ~45 kDa proteins [13]. The protein deglutathionylation activities of WT and cysteine mutants were measured using preadipocyte cell proteins as the substrate (Table 1). The activity of WT was determined to be 3.0 mU/mg, whereas C32A had no detectable activity. Other cysteine mutants exhibited deglutathionylation activities similar or slightly higher than WT. Protein deglutathionylation was confirmed by Western blot analysis. WT consistently exhibited deglutathionylation activity, showing ~45 kDa glutathionylated proteins (Pro-SSG) at a lower level than that of the control without GstO1 (Figure 6). C32A did not show deglutathionylation, with no change in Pro-SSG. While other cysteine mutants showed deglutathionylation of Pro-SSG, C236A was the most significant.

## 3. Discussion

GstO1 accepts various substrates in vitro and catalyzes their reduction or deglutathionylation. But, in cells, glutathionylated proteins were found and identified to be the physiologically relevant substrates of GstO1 for deglutathionylation [6]. Protein glutathionylation is a reversible post-translational modification that plays a crucial role in protecting proteins from irreversible oxidative damage under conditions of oxidative stress. GstO1 catalyzes protein deglutathionylation and therefore regulates the structures and functions of glutathionylated proteins. The dysregulation of GstO1-mediated deglutathionylation has been implicated in various diseases associated with oxidative stress and protein dysfunction [1,7].

Oxidative stress occurs when there is an imbalance between reactive oxygen species (ROS) and antioxidants in cells, such as H_2_O_2_ and GSH, respectively. In this study, we found that GstO1 is sensitive to H_2_O_2_, which destabilizes the protein, causing denaturation and precipitation at a physiological temperature. In contrast, GSH, the prevalent antioxidant in cells, effectively prevented the H_2_O_2_-induced destabilization. Furthermore, conserved cysteine residues C32 and C236 in GstO1 have been identified as critical for sensing changes in the redox environment, which explain the pathological cysteine mutations in patients with associated diseases.

The catalytic cysteine mutant C32A exhibited a dramatic decrease in thermal stability compared to WT (Figure 3 and Table 2), indicating that C32 is a critical residue, not only for the catalytic activity, but also for the stability of GstO1. In the presence of H_2_O_2_, C32A was highly unstable and rapidly denatured at physiological temperature but was stabilized by GSH (Figure 4). However, the stabilizing effect of GSH on C32A was much less than that observed for WT, possibly due to the lack of a disulfide bond between GSH and C32 [5,8]. Other conserved cysteine residues are also important for the stability of GstO1, as shown by changes in the thermal stabilities of cysteine mutants. In particular, C236A was highly stable under various redox conditions, even in the presence of H_2_O_2_, which dramatically reduced the stability of WT and other cysteine mutants. These results indicate that C236 is critically required to regulate the GstO1 stability, and to sense oxidizing environments. Cysteine mutants adopt conformations different from the WT conformation (Figure 1). Their conformations also vary depending on the redox environment (Figure 5). This redox-dependent conformational change was least pronounced for C236A, which concurred with the minimal change in stability at physiological temperature. These results indicate that the stability of GstO1 is affected by changes in the redox environment, and that C32 and C236 play pivotal roles by sensing redox changes.

As expected, the mutation of the catalytic residue C32 abolished the enzyme activities of GstO1 (Table 1). But, a low glutathione *S*-transferase activity was detected for an unknown reason, consistent with the previous report [8]. Other cysteine mutants C90A and C191A showed slight changes in enzyme activities. However, C236A exhibited significant differences compared to WT, showing higher thiol transferase, deglutathionylation, and DHA reductase activities, but a lower 4-NPG reductase activity. In particular, the protein deglutathionylation activities of C90A, C191A, and C236 were determined to be higher than the WT activity, despite the limited availability of glutathionylated protein substrates in total cell proteins. Glutathionylated proteins of ~40 kDa were detected at significant levels in preadipocytes, consistent with the previous report [13]. WT reduced levels of glutathionylated proteins, which was absent with C32A (Figure 6). Other cysteine mutants showed lower levels of glutathionylated proteins than WT, confirming their higher deglutathionylation activities. The C32A mutant rendered the protein inactive for protein deglutathionylation, as expected, since the C32 is the catalytic residue. However, other conserved cysteine mutants exhibited higher deglutathionylation activities compared to WT. Notably, C236A showed the most significant increase in the protein deglutathionylation activity. These results suggest that C236 plays a crucial role in the protein structure by sensing redox environments such as H_2_O_2_, while also affecting the protein deglutathionylation activity of GstO1. Elucidating the structure of C236A could reveal the critical role of C236 and its pathological implications.

Oxidative stress (H_2_O_2_) and antioxidants are well-known regulators during adipocyte differentiation via different signaling pathways [18,19]. Although the regulation mechanism has not been fully elucidated, intracellular H_2_O_2_ levels are crucial, which enhances differentiation at lower levels but suppresses differentiation at higher levels. Several proteins are regulated by glutathionylation, including β-actin and C/EBP-β, which play critical regulatory roles in adipocyte differentiation [20,21,22,23]. Previously, we reported that the inhibition of the deglutathionylation activity of GstO1 effectively suppresses adipocyte differentiation [13]. Building on these previous findings, the results in this study strongly indicate that GstO1 can mediate the redox-dependent regulation of adipocyte differentiation by sensing the redox environment and regulating the deglutathionylation of specific adipogenic proteins. Furthermore, this study reveals that conserved cysteine residues C32 and C236 are critical for sensing changes in the redox environment to regulate the stability and function of GstO1, which explains pathological cysteine mutations found in patients with associated diseases.

## 4. Materials and Methods

### 4.1. Protein Preparation

Wild-type and mutant GstO1 were prepared as N-terminal 6× His-tagged proteins by over-expression of the encoding genes in *E*. *coli* BL21 and Ni-NTA affinity protein purification as previously described [13]. Each conserved cysteine residue of GstO1 was mutated to alanine by a site-directed mutagenesis PCR using primers (Appendix A).

### 4.2. Enzyme Activity Assays

The activities of thiol transferase, dehydroascorbate reductase, glutathione *S*-transferase, and 4-nitrophenacyl glutathione (4-NPG) reductase were measured as previously described [6,13]. Protein deglutathionylation was measured using 3T3-L1 preadipocyte proteins. Cells were grown in the media DMEM/10% calf serum to confluency, and it was allowed another 2 days to reach growth arrest. Cell proteins were extracted by lysing harvested cells in RIPA (Thermo scientific, Waltham, MA, USA), and protein concentrations were determined by BCA assay (Thermo scientific). The deglutathionylation assay contained 100 μg/mL of cell proteins in 50 mM Tris/HCl pH 8.0, 150 mM NaCl, 0.2 mM NADPH, 1.0 mM EDTA, 0.1 mM GSH, and 2 units of glutathione reductase. The deglutathionylation reaction was initiated by adding GstO1, followed by measuring the absorbance decreases at 340 nm. Deglutathionylation activity was calculated using an extinction coefficient of 6.2 mM^−1^·cm^−1^.

### 4.3. Thermal Stability Assays

The thermal stability of GstO1 was determined at different temperatures by differential scanning fluorimetry (DSF), and at constant temperature by isothermal denaturation (ITD), as previously described in [13,24]. The assay mixture contained 0.1 mg/mL of GstO1 in 50 mM Tris/HCl pH 8.0, 150 mM NaCl, 10× Sypro orange (Invitrogen, San Diego, CA, USA), the indicated concentrations of H_2_O_2_, and GSH. The assay mixture was incubated by increasing the temperature (1 °C/min) for DSF or was incubated at the constant temperature of 37 °C for ITD. Protein denaturation was followed by measuring fluorescence increases using an ABI 7500 Real-Time PCR system (Applied Biosystems, Foster City, CA, USA). The temperature at half-maximal denaturation (*T*_m_) was determined by fitting DSF denaturation curves to the Boltzmann equation. The protein denaturation rate (*k*_ob_) was determined by fitting a denaturation curve to a single exponential equation. The H_2_O_2_ concentration for the half-maximal denaturation rate (EC_50_) was determined by the plot of (Δ*k*_ob_ versus [H_2_O_2_], Δ*k*_ob_ = *k*_ob_ at the indicated H_2_O_2_—*k*_ob_ at no H_2_O_2_) to a hyperbolic equation of Δ*k*_ob_ = maxΔ*k*_ob_ × [H_2_O_2_]/(EC_50_ + [H_2_O_2_]).

### 4.4. Western Blot Analysis

Cell proteins in protein deglutathionylation assays were separated by 12% SDS-PAGE under non-reducing conditions for Pro-SSG, and under reducing conditions for β-actin. Separated proteins were transferred by a PVDF membrane and probed using an anti-Pro-SG antibody (Merck, Darmstadt, Germany) or an anti-β-actin antibody (Santacruz, Santa Cruz, CA, USA).

### 4.5. Statistical Analysis

The significance of differences in enzyme activities (Table 1 and Figure 6B) and thermal stabilities (Table 2) between WT and mutant GstO1 was determined using the Student’s *t*-test for unpaired samples and one-way ANOVA. Results are presented as means ± standard errors, and a *p*-value < 0.01 was considered statistically significant.

## Figures and Tables

**Figure 2 ijms-25-05279-f002:**
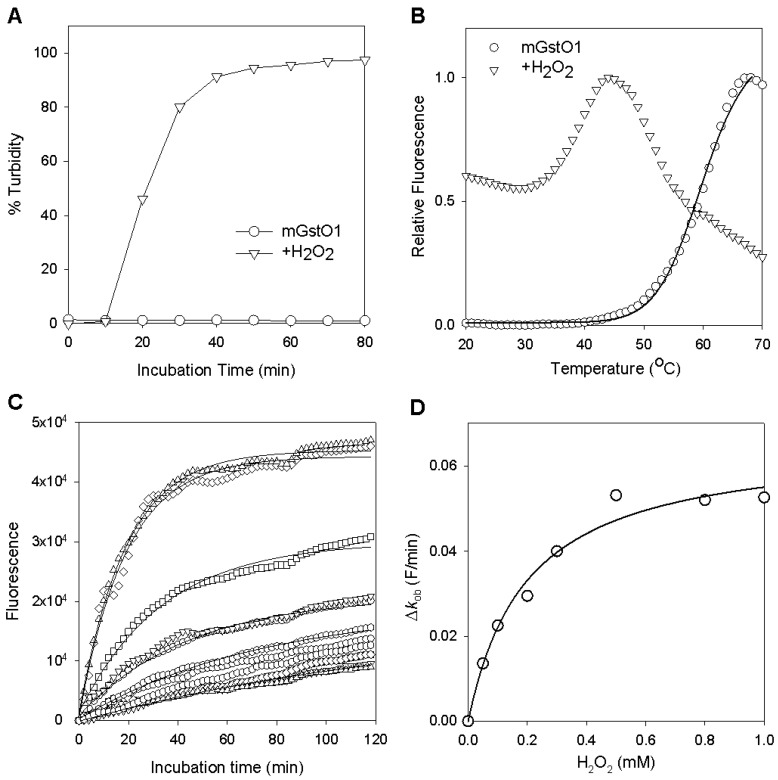
Hydrogen peroxide-induced destabilization of GstO1. (**A**) Precipitation of WT GstO1 in the absence or presence of 1.0 mM H_2_O_2_. Turbidity was measured at 700 nm at room temperature. (**B**) Thermal stability of WT GstO1 was measured by DSF in the absence or presence of 1.0 mM H_2_O_2_. The solid line is a fitting of the denaturation curve to a Boltzmann equation. (**C**) Thermal stability of WT GstO1 measured by ITD at 37 °C in the presence of 0–1.0 mM H_2_O_2_. The solid lines are fits of denaturation curves to single exponential equations. (**D**) Plots of ITD denaturation rates versus H_2_O_2_ concentrations. The solid line is a fit to a hyperbolic equation to determine the H_2_O_2_ concentration for the half-maximal denaturation rate (EC_50_), as described in the materials and methods.

**Figure 3 ijms-25-05279-f003:**
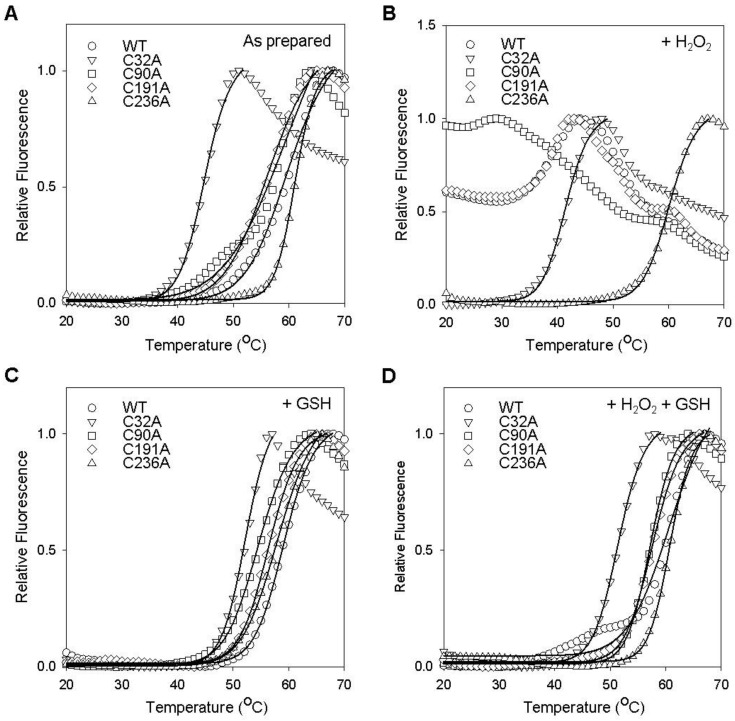
Redox-dependent changes in thermal stability on increasing temperature. (**A**–**D**) Thermal stabilities of WT and mutant GstO1 measured by DSF in the absence (**A**) and presence of 1.0 mM H_2_O_2_ (**B**), 1.0 mM GSH (**C**), or 1.0 mM H_2_O_2_ + 1.0 mM GSH (**D**). The solid lines are the fits of the denaturation curve to Boltzmann equations, and the determined *T*_m_ ± SD (n > 3, °C) are summarized in Table 2.

**Figure 4 ijms-25-05279-f004:**
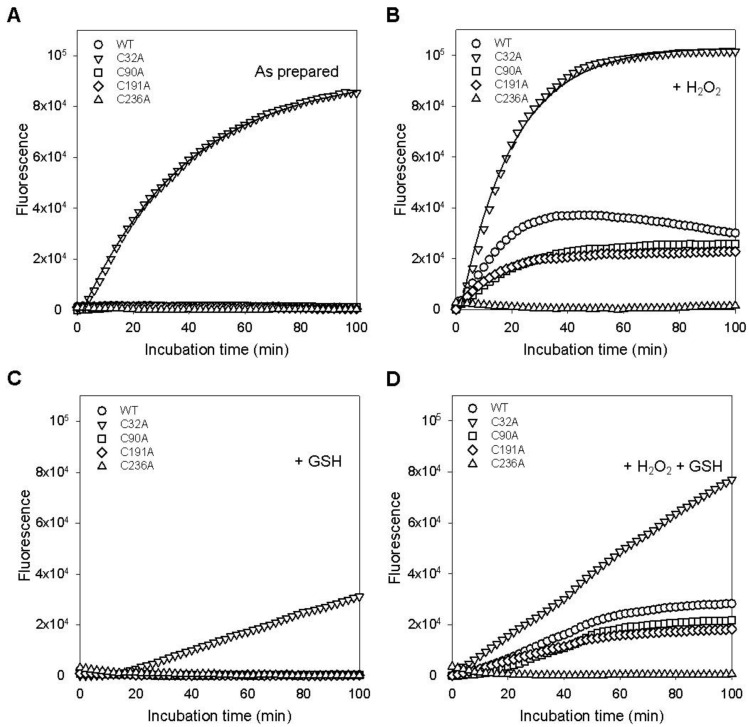
Redox-dependent changes in thermal stability at a constant temperature. (**A**–**D**) Thermal stabilities of WT and mutant GstO1 measured by ITD at 37 °C in the absence (**A**) and presence of 1.0 mM H_2_O_2_ (**B**), 1.0 mM GSH (**C**), or 1.0 mM H_2_O_2_ + 1.0 mM GSH (**D**). Solid lines are the fits of denaturation curves to single exponential equations.

**Figure 5 ijms-25-05279-f005:**
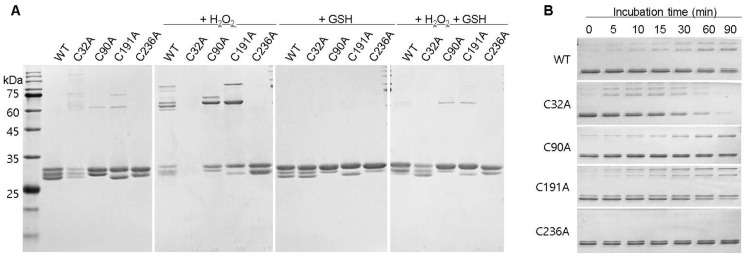
Redox-dependent changes in conformation. (**A**) Non-reducing SDS-PAGE analysis after the 2 h incubation of WT and mutant GstO1 at 37 °C with indicated 1.0 mM H_2_O_2_ or (and) 1.0 mM GSH. (**B**) Non-reducing SDS-PAGE analysis for WT and mutant GstO1 at the indicated incubation times.

**Figure 6 ijms-25-05279-f006:**
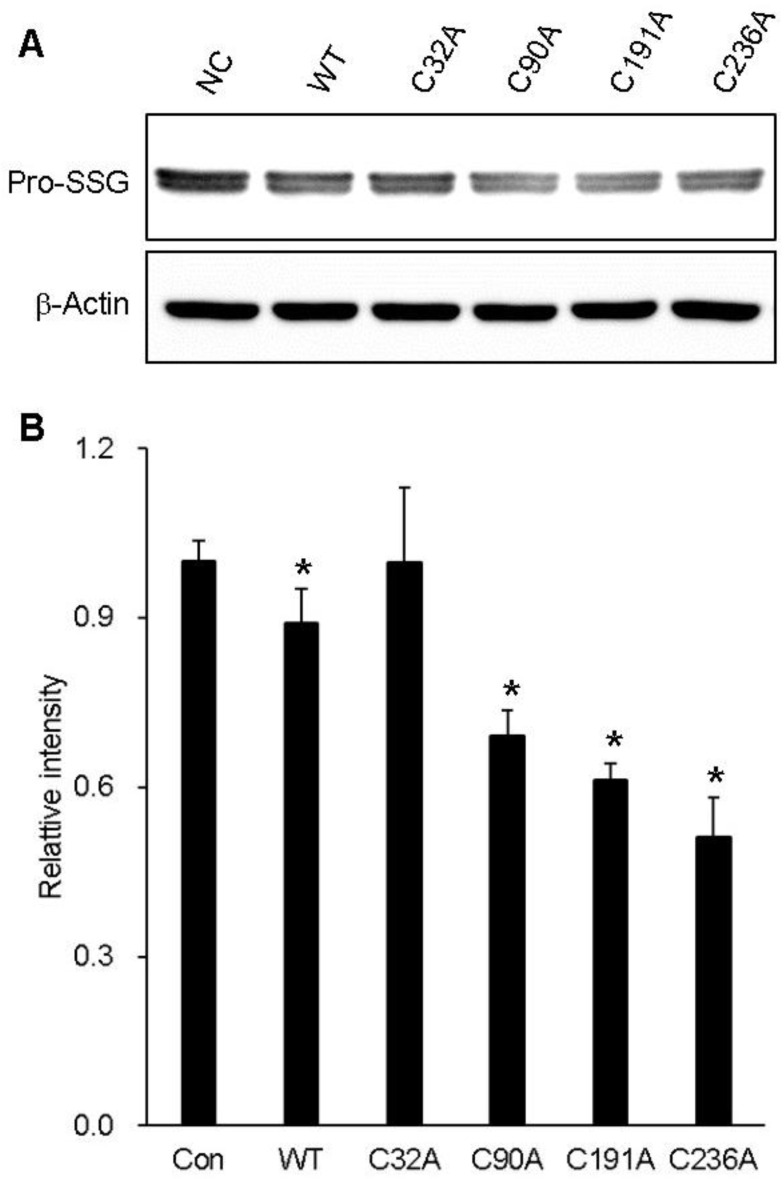
Deglutathionylation of cell proteins. (**A**) Pro-SSG analyzed by Western blot for 3T3-L1 cell proteins after incubation with WT and mutant GstO1. (**B**) Quantitative analysis of Pro-SSG normalized by β-actin using Image J software version 1.54 g (n ≥ 3) with *p* values < 0.05 (*) are indicated.

**Table 1 ijms-25-05279-t001:** Enzyme activities of GstO1.

	Thiol Transferase (mU/mg)	Deglutathionylation (mU/mg)	DHA Reductase (mU/mg)	Glutathione *S*-Transferase (mU/mg)	4-NPG Reductase (U/mg)
WT	116 ± 11	3.0 ± 0.1	132 ± 31	ND	9.1 ± 0.2
C32A	ND	ND	ND	40 ± 4	ND
C90A	101 ± 18	3.1 ± 0.1	133 ± 31	ND	10.4 ± 0.6
C191A	111 ± 13	3.1 ± 0.1	146 ± 10	ND	10.6 ± 0.8
C236A	150 ± 22	3.5 ± 0.1	166 ± 19	ND	6.1 ± 0.6

ND, not detected (n ≥ 5).

**Table 2 ijms-25-05279-t002:** Thermal stability (*T*_m_) of GstO1 determined by DSF.

	Control	H_2_O_2_	GSH	H_2_O_2_ + GSH
WT	60.0 ± 0.3	<20 *	59.0 ± 0.2	61.0 ± 0.8
C32A	45.0 ± 0.3	41.0 ± 0.8	52.0 ± 0.5	51.0 ± 0.8
C90A	57.0 ± 1.5	<20 *	54.0 ± 0.7	57.0 ± 0.4
C191A	56.0 ± 0.7	<20 *	56.0 ± 0.6	58.0 ± 0.7
C236A	61.0 ± 0.2	60.0 ± 0.4	58.0 ± 0.4	61.0 ± 0.8

*, *T*_m_ < 20 °C were estimated as protein denaturation at room temperature (n ≥ 5).

## Data Availability

All data is contained within the article and Appendix A.

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
