# Peer review of "Regulation of Glutathione S-Transferase Omega 1 Mediated by Cysteine Residues Sensing the Redox Environment"

_ijms, 2024, doi:10.3390/ijms25105279_

Round 1

Reviewer 1 Report

Comments and Suggestions for Authors

In this study, the authors investigated the effects of cysteine mutations on the structure and function of glutathione transferase omega 1 under different redox conditions. Unlike other GST, class omega has different activities such as thioltransferase, DHAR and is involved in arsenic metabolism due to specific amino acid in the active site, different from other GSTs. Still, a lot about GSTO1 and GSTO2 is unknown and the topic of this article is original and relevant in the field. To my knowledge, there is no published paper regarding the effects of cysteine mutations on the structure and function of glutathione transferase omega 1 under different redox conditions. The methodology used is adequate for the carried research. It would be interesting to see the results of the same research repeated in specific human tissue, malignant or non-malignant. The conclusions are consistent with the evidence and arguments presented and they address the main hypothesis. The references are appropriate, there are two more articles that give additional aspects of role of GSTO2 in diseases:

Airways glutathione S-transferase omega-1 and its A140D polymorphism are associated with severity of inflammation and respiratory dysfunction in cystic fibrosis. J Cyst Fibros. 2021 Nov;20(6):1053-1061. doi: 10.1016/j.jcf.2021.01.010.

Upregulated glutathione transferase omega-1 correlates with progression of urinary bladder carcinoma. Redox Rep. 2017 Nov;22(6):486-492. doi: 10.1080/13510002.2017.1299909.

Is it possible for the authors to provide a model figure of how cysteine mutations affect the structure and function of glutathione transferase omega 1 under different redox conditions (like some kind of graphical abstract, not necessary if they find it too inconvenient).

Reviewer 2 Report

Comments and Suggestions for Authors

In this manuscript, Kim et al. investigated effects of cysteine mutations, and redox conditions on the function of the glutathione S-transferase Omega-1 (GstO1). The authors demonstrated that (1) cysteine mutations of GstO1 alters its biochemical properties and enzymatic activity, (2) Redox conditions, specifically with the addition of hydrogen peroxide, affects GstO1 stability. (3) Conserved cysteines are critical for GstO1 conformation, stability under oxidizing conditions, and function.

The manuscript is of good quality, and presents some interesting and novel data, with some potential pitfalls. I recommend that this manuscript is suitable to be published after the revision suggested below:

(1)     The manuscript presented large amounts of data, but lacked proper discussion and explanation of the data. For example, in Table 1, authors presented enzymatic activity data but did not propose a possible underlying reason behind enzymatic activiety differences. Having more of the discussions on the data rather than simply presenting them will help reader grasp better and provide more insight on the data.

(2)     Could authors provide an explanation and clarification on the bump in fluorescence signal around 40°C for H2O2  in Fig. 2B?

(3)     In Fig. 2D, could authors provide justification for fitting an IC50 curve using hyperbolic equation?

(4)     For section 2.5, it is intriguing that C236A mutant exhibits more significant effect than the catalytically dead C32A. The authors may want to provide more discussion as this topic is worth much deeper future investigation.

Minor points:

(1)     It appears that no statistical tests were preformed on data presented in tables 1 and 2. The authors need to perform the tests if they claim that the results are significant/insignificant, e.g. in section 2.3.

(2)     In line 159, “highly stable” is confusing statement, perhaps using words such as “resistant to H2O2 denaturation” would be better.

(3)     In line 250, the paper referenced may be a mistake because the reference did not discuss about the purification of heterologous expressed GstO1.

(4)     In line 283, please clarify “gene expression levels”, since the statistical tests performed in this present paper is quantification of Pro-SSG western blot, the author will need to clarify what they meant by “gene expression levels.
